# Updating the description of *Rhizobium* diversity associated with common bean cultivars in the Ecuadorian Andes: A phylogenetic and functional perspective

Andrea León–Cadena[1]*, Henry D. Naranjo[2,3]*, Janine Jiménez–Parra[4], José Ochoa[5,6], Michelle Avalos–Loayza[1,7], Pamela Murillo[1], Angel Murillo[8], Gustavo Bernal[9], Juan Cadena–Villota[1,10], Lenin Ron-Garrido[1,3,5,10]*

**1** Instituto de Investigación en Etnociencias, Universidad Central del Ecuador, Quito, Ecuador, **2** Facultad de Ciencias Químicas, Universidad Central del Ecuador, Quito, Ecuador, **3** Instituto de Investigación en Zoonosis (CIZ), Universidad Central del Ecuador, Quito, Ecuador, **4** Unidad de Recursos Naturales, Centro de Investigación Científica de Yucatán, Mérida, México, **5** Facultad de Ciencias Agrícolas, Universidad Central del Ecuador, Quito, Ecuador, **6** Departamento de Protección Vegetal, Instituto Nacional de Investigaciones Agropecuarias (INIAP), Quito, Ecuador, **7** Programa Nacional de Leguminosas y Granos Andinos, Instituto Nacional de Investigaciones Agropecuarias (INIAP), Quito, Ecuador, **8** Facultad de Ciencias, Universidad Central del Ecuador, Quito, Ecuador, **9** Independent Researcher, Quito, Ecuador, **10** Facultad de Medicina Veterinaria y Zootecnia, Universidad Central del Ecuador, Quito, Ecuador

☯ These authors contributed equally to this work.
* alleon@uce.edu.ec (ALC); hdnaranjo@uce.edu.ec (HN); ljron@uce.edu.ec (LRG)

## Abstract

*Phaseolus vulgaris* (common bean) is nodulated by diverse *Rhizobium* species. Although Ecuador is recognized as one of the centers of bean diversification, its native rhizobial diversity and geographic distribution remains poorly characterized. We isolated 46 native *Rhizobium* strains from root nodules across four Andean provinces (Imbabura, Pichincha, Chimborazo, and Loja). Partial sequencing of the *recA* gene delineated nine strain clusters (R1–R9) within two major phylogenetic groups: (i) *Rhizobium ecuadorense/Rhizobium leguminosarum/Rhizobium etli/Rhizobium phaseoli* and (ii) *R. tropici*. Multilocus sequence analysis (MLSA) of the housekeeping genes *recA*, *glnII*, *dnaK* genes from 19 representative isolates showed four phylogenetic clusters (C1–C4). Cluster C1 (*R. ecuadorense*–related) predominated in northern Ecuador; C2 formed a distinct Chimborazo cluster; C3 appeared sporadically in Imbabura and Chimborazo; and C4 (*R. tropici*–related) was confined to Loja's Amotape–Huancabamba Zone and displayed unique phenotypes. In greenhouse assays on two local bean varieties, all isolates formed nodules in both varieties; several isolates induced significantly higher nodule counts than the commercial inoculant UMR1899 (*Rhizobium tropici* IIB CIAT 899^T). These results suggest geographic variation among Ecuadorian *Rhizobium* populations and identify locally predominant groups for further evaluation as bioinoculants.

**Data availability statement:** All relevant data are within the paper and its Supporting information files.

**Funding:** This study was supported by the Universidad Central del Ecuador through the Convocatoria de Proyectos de Investigación Senior UCE-2019 (DI-CONV-2019-033). Additional support in logistics and laboratory infrastructure was provided by the Instituto de Investigación en Etnociencias and the Instituto de Investigación en Zoonosis (CIZ). The funders had no role in study design, data collection and analysis, decision to publish, or preparation of the manuscript.

**Competing interests:** The authors have declared that no competing interests exist.

## Introduction

Nitrogen is an essential component of biological systems, serving as integral component of proteins, nucleic acids, and other vital organic molecules [1]. Although nitrogen ($N_2$) makes up 78% of the atmosphere, converting it into usable forms like ammonia ($NH_3$) requires significant energy and occurs either through the industrial Haber–Bosch process or naturally via biological nitrogen fixation (BNF) by microorganisms [2,3]. The majority of BNF is carried out by diazotrophs, bacteria and archaea that live freely or in symbiosis with plants, catalyzing the conversion of atmospheric nitrogen ($N_2$) into ammonia via the nitrogenase enzyme, which requires large amounts of ATP [4].

Among the most important symbiotic nitrogen–fixing bacteria are the rhizobia, a group of soil–borne bacteria belonging to various genera within the Proteobacteria [5]. These bacteria establish a symbiotic relationship with legumes, infecting their roots and leading to the formation of nodules. Nitrogen is a key limiting factor for crop yields, and BNF plays a critical role in restoring soil fertility in agricultural systems [6,7]. Consequently, rhizobia play a key role in this process by reducing dependence on synthetic fertilizers, which frequently leach into and damage sensitive ecosystems [8,9].

The common bean (*Phaseolus vulgaris*), a legume native to the Americas, is among the most important grain legumes globally, especially valued for its high protein, fiber, and essential mineral content. It is particularly significant for small–scale farmers across Latin America and Sub–Saharan Africa, where it serves as a vital source of nutrition and income [10–13]. Like other legume species, common beans form root–nodule symbiosis mainly with *Rhizobium* spp., and less frequently with *Ensifer, Pararhizobium*, *Bradyrhizobium* and *Burkholderia* [14,15]. The genus *Rhizobium*, part of the class Alphaproteobacteria, comprises more than 40 symbiotic species based on *16S rRNA* analysis [12,16–18]. In the Americas, particularly in the domestication and diversification centers of *P. vulgaris* (Mesoamerica and Andes), *Rhizobium etli* and the related lineage *Rhizobium phaseoli* are the predominant microsymbionts. Nevertheless, other *Rhizobium* species, such as *Rhizobium leguminosarum, Rhizobium tropici, Rhizobium ecuadorense*, and *Rhizobium gallicum*, have also been identified as symbionts in these regions [17,19–27]. It is worth noting that strains isolated from the northern and central Andean bean cultivation areas of Ecuador, as described by Bernal and Graham [26] and later characterized by Ribeiro *et al*. [21], were identified as a new lineage that likely predominates in these regions. Further analysis by Ribeiro *et al*. [28] confirmed that these strains represented a novel species within the genus *Rhizobium*, described as *Rhizobium ecuadorense* CPSo 671ᵀ.

More than 20 years have passed since Bernal and Graham [26] conducted the first study on the diversity of rhizobia nodulating *P. vulgaris* in Ecuador [26]. Despite significant advances in sequencing techniques and genetic characterization of microorganisms that allow us to achieve a high level of precision in taxonomic description, only a few updates have been published in Ecuador on this subject in the last two decades. This is particularly concerning given that the Ecuador–Peru region, situated

between two major centers of *P. vulgaris* diversification, constitutes a unique genetic reservoir, making it an ideal location for investigating the coevolutionary dynamics between common bean and its symbiotic rhizobia [10,29].

In this context, this study aimed to characterize diverse *Rhizobium* isolates obtained from *P. vulgaris* root nodules collected across four Ecuadorian Andean provinces: Imbabura, Pichincha, Chimborazo and Loja. Phylogenetic affiliations were first established using *recA* gene analysis, followed by *16S rRNA* sequencing and multilocus sequence analysis (MLSA) of concatenated housekeeping genes *recA, glnII and dnaK* to achieve higher phylogenetic resolution. Additionally, the nodulation capacity and symbiotic efficiency of native isolates were assessed in greenhouse trials using two commercial bush–bean varieties (Canario and Centenario).

## Materials and methods

### Sampling locations and rhizobia isolation from root nodules

Root nodules of *P. vulgaris* were systematically collected from agricultural soils across the Ecuadorian provinces mentioned above during 2019–2021. Sampling was conducted at twenty different locations, under research permits Nro. MAE-DNB-CM-2015–0028, Nro. MAE-DNB-CM-2015–0028-M-0001 and Nro. MAE-DNB-CM-2015–0028-M-003, issued by the Ministerio del Ambiente del Ecuador. Geographic coordinates of each site are provided in Fig 1 and S1 Table.

Up to three common bean plants were randomly collected from each site. The roots and their nodules were carefully separated from the plants, wrapped in sterile moist towels, and placed in sealed plastic bags. These samples were then transported in a cooler to the laboratory of the Instituto de Investigación en Zoonosis at Universidad Central del Ecuador, and stored at 4 °C to maintain their integrity until further processing [31].

Pink root nodules randomly selected from each sample were surface–sterilized according to the procedure described by Hungria et al. [32]. Briefly: the nodules, placed inside a sterile tea strainer, were successively submerged in 70% (v/v) ethanol for 1 min, 4% (v/v) sodium hypochlorite for 3 min, and finally rinsed six times in sterile deionized water. After disinfection, the nodules were crushed using sterile blunt–tip tweezers, and the released contents were plated on yeast extract–mannitol agar (YMA) [33] supplemented with 0.25% Congo red. The plates were then incubated at 28 °C for three to seven days. Colonies were subcultured on YMA supplemented with 0.5% of bromothymol blue (BTB) indicator to assess acid/alkali reaction and subsequently re–plate on YMA to obtain pure cultures. The pure cultures were preserved in 15% (v/v) glycerol–Yeast Mannitol Broth (YMB) at −80 °C [26,34]. Tolerance to NaCl (1% and 2% w/v) was evaluated on YMA at 28 °C, and growth at 37 °C and 40 °C was also assessed on YMA (S2 Table).

### Preliminary phylogenetic analysis of rhizobia isolates via *recA* allele amplification

Forty–five bacterial isolates preserved in glycerol, and two reference strains UMR1899 (*Rhizobium tropici* IIB CIAT 899[T]) and UMR1632 (*Rhizobium etli* CIAT 632), were streaked on YMA plates and incubated at 28 °C for 2–5 days. Individual colonies from each isolate were subsequently re–plated on fresh YMA plates, forming a single line at a designated position on the plate, with up to four isolates per plate. The fast–growing isolates were incubated for 16 hours and the slow–growing isolates for 20 hours, at 28 °C. Colony PCR targeting the partial *recA* gene was conducted to evaluate strain diversity using primers and thermal cycling conditions outlined in Supplementary S3 Table. The PCR products were verified by electrophoresis on a 1.2% agarose gel at 100 V for 40 minutes, with an ABM 100 bp Optic–DNA Marker included as a reference. The gel was stained with SYBR™ Safe DNA Gel Stain and visualized under ultraviolet (UV) light using a trans–illuminator (LABNET) for the detection of amplicons. High–quality PCR amplicons were sent to Macrogen Inc. (Seoul, Korea) for purification and Sanger sequencing.

The resulting *recA* gene sequences were analyzed and edited manually using MEGA X software [35]. The acquired sequences were subsequently searched using Basic Local Alignment Search Tool (BLASTN) on the National Center for Biotechnology Information (NCBI) server (http://www.ncbi.nlm.nih.gov/blast) to identify closely related type sequences. Both sets of sequences were exported and aligned using ClustalW software within MEGA X [35]. A phylogenetic tree was

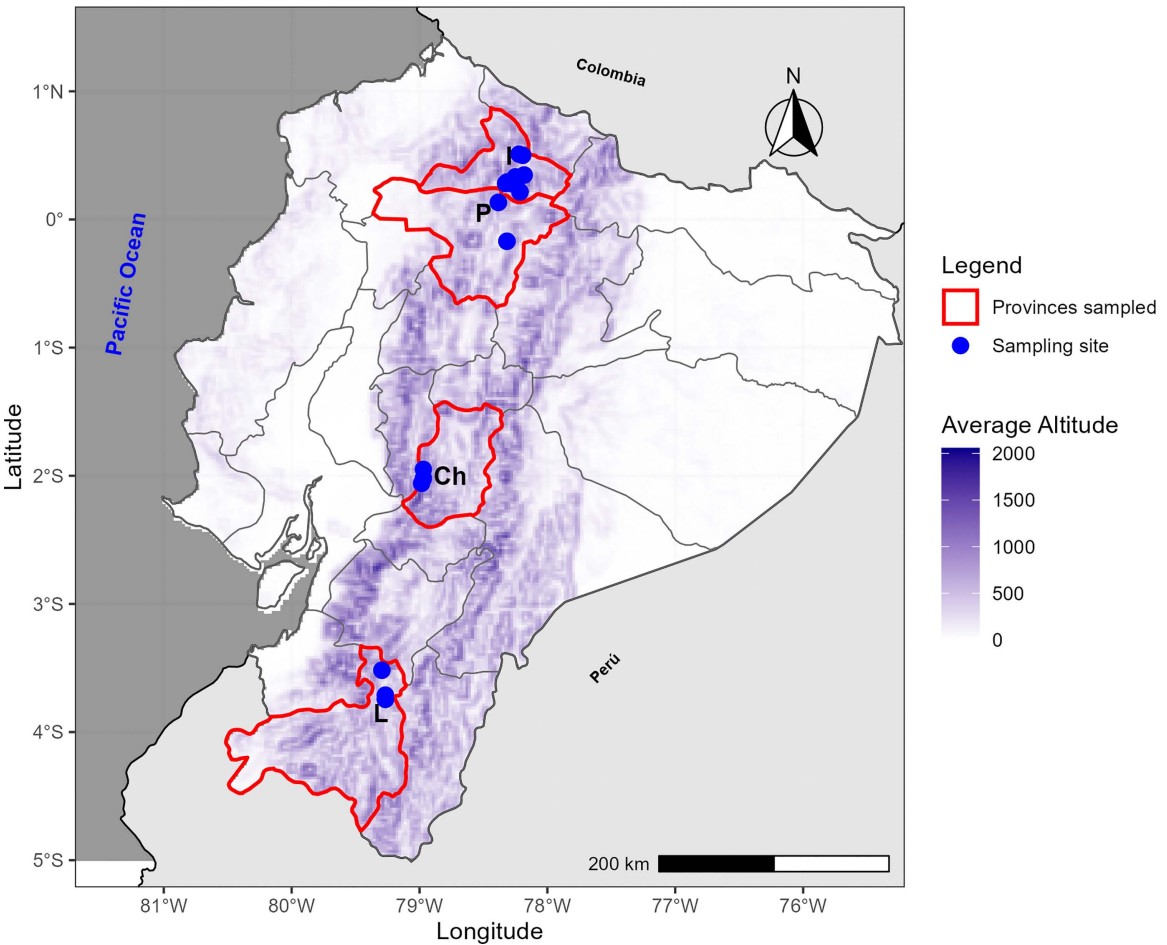

**Fig 1. Map of Ecuador indicating the geographic locations of sampling sites in Imbabura (I), Pichincha (P), Chimborazo (Ch), and Loja (L) Provinces.** Further details regarding these sampling areas are provided in S1 Table. Basemap and administrative boundary data from Natural Earth (public domain). Elevation data from open-access DEM sources accessed via the elevatr package. Figure created entirely by the authors in R using sf, raster, and ggplot2 [30]. These data are compatible with publication under the CC BY 4.0 license from PLOS ONE.

constructed using the Maximum Likelihood (ML) method, with the optimal model selected based on Bayesian Information Criterion (BIC) values. Bootstrap analysis was conducted with 1000 replicates; bootstrap values above 70% are shown at nodes. Pairwise similarity (%) of *recA* genes was estimated using the distance module in MEGA X. Each isolate was designated with the acronym UCE followed by a four–digit code and a letter indicating its province of isolation (**I**=Imbabura; **P**=Pichincha; **Ch**=Chimborazo; **L**=Loja).

The green bars indicate the altitude range where samples were collected, spanning from 1200 to 2814 meters above sea level. Isolates were also assessed for salt tolerance (indicated by filled stars) and temperature tolerance (indicated by filled circles). Type strains were indicated with superscript "T" following the strain code.

### Amplification and sequencing of 16S rRNA gene and housekeeping genes *recA, glnII* and *dnaK*

Based on the *recA*–based phylogenetic analysis described above, 19 representative isolates were selected from distinct clusters according to their collection sites and phylogenetic grouping (S2 Table). Additionally, the reference strains UMR1899 and UMR1632 were incorporated into the analysis. To further resolve the taxonomic position of these isolates,

we performed *16S rRNA* sequence analysis followed by MLSA using concatenated sequences of the housekeeping genes *recA*, *glnII*, and *dnaK*.

The 19 representative rhizobial isolates were inoculated into 5 mL of YMB medium and incubated at 30 °C in a shaking incubator at 150 rpm until reaching the early stationary phase. Genomic DNA was extracted using a GENEJET GENOMIC DNA PURIFICATION KIT according to the manufacturer's specifications. The quality and concentration of purified DNA were determined using a NanoDrop 2000 spectrophotometer (Thermo Scientific, Waltham, USA) and by electrophoresis on a 1% (w/v) agarose gel. High–purity DNA was stored at −20 °C and subsequently used as a template for PCR amplification of the *16S rRNA*, *recA*, *glnII* and *dnaK* genes. The amplification procedures, including primer sequences and protocols, are detailed in Supplementary S3 Table.

PCR products were purified using the ABM Column−Pure Gel and PCR Clean−up kit following the manufacturer's instructions. The purified DNA fragments were sent to Macrogen Inc. (Seoul, Korea) for Sanger sequencing. The sequences were manually edited and assembled with the MEGA X software package. Partial 16S rRNA gene sequences were used to search for bacterial–type strains that showed high similarity (≥ 98.65%) in EzTaxon−e [36,37]. The *16S rRNA* sequences of isolates, along with those of type strains, were subsequently imported into MEGA X to construct a phylogenetic tree using the Maximum Likelihood method and Tamura−Nei model. Cluster support was assessed through bootstrap analysis with 1000 replicates. All gene sequences were deposited in the NCBI database, and the corresponding accession numbers are listed in the Supplementary S2 Table.

Genes, *recA, glnII* and *dnaK* were analyzed with BLASTN (Basic Local Alignment Search Tool) using the NCBI server to find closely related reference sequences. These housekeeping genes were aligned using ClustalW in Mega X and appropriately trimmed. Subsequently, the *recA*, *glnII* and *dnaK* genes were concatenated to construct a phylogeny inferred using the Maximum Likelihood method and General Time Reversible model in MEGA X [35,38]. Cluster support was assessed through bootstrap analysis with 1000 replicates. Pairwise genetic distances were calculated using MEGA X. Finally, all sequences were deposited in the NCBI database, with accession numbers provided in supplementary information S2 Table.

## Plant nodulation authentication test

The nodulation capacity was evaluated by inoculating 42 of the 46 rhizobial isolates (excluding four from Chimborazo due to post–pandemic logistical constraints), along with two reference strains UMR1899 and UMR1632 supplied by the United States Department of Agriculture (USDA). Two commercial varieties of *P. vulgaris* (INIAP–420 Canario and INIAP–484 Centenario) were used in the inoculation trials, which were conducted under greenhouse conditions at the Instituto Nacional de Investigaciones Agropecuarias (INIAP) in Tumbaco *(0°12′54″ S; 78°24′04″ W)*. Isolates from Chimborazo province (UCE0221, UCE0224, UCE0228, and UCE0231) were collected after the COVID–19 pandemic due to mobility constraints and after completion of the greenhouse trials; therefore, the trial could not be repeated. However, molecular characterization of the *recA* gene was completed.

For the inoculation assay, bean seeds were surface–sterilized and pre–germinated on 2% agar plates at 28 °C for three days [39]. Pre–germinated bean seeds were individually transplanted into 38.1 cm × 50.8 cm free–draining polyethylene bags filled with 0.5 kg of steam–sterilized vermiculite as a substrate. Rhizobial strains were grown on agar plates for 2–5 days, suspended in sterile deionized water, and adjusted to a final concentration of $10^9$ CFU mL$^{-1}$. Seedlings were inoculated with 1 mL of the prepared suspension.

Each treatment included three technical replicates, with one seedling per bag. Bags received 150 mL of sterile nutrient solution three times a week [40]. Controls consisted of non–inoculated plants grown with or without nitrogen supplementation for each bean variety [39]. At 45 days post–inoculation, plants were harvested by cutting the shoot at the substrate surface, and four variables related to nodulation capacity were measured: (i) total number of nodules per root system, (ii) number of nodules >2 mm in diameter iii) number leghemoglobin–positive nodules >2 mm; and (iii) percentage of leghemoglobin–positive nodules.

The mean and standard deviation of the evaluated variables were calculated in RStudio (version 4.1.0), and all values were standardized to eliminate scale differences, ensuring that each variable had a mean of 0 and a standard deviation of 1. Here, we present nodulation data for 15 isolates with available MLSA molecular data; reference strains were excluded. A dendrogram based on geographic distances was constructed to illustrate the clustering of native isolates from Pichincha, Imbabura, and Loja. Additionally, a heat map of the standardized variables was generated to assess whether geographical location correlates with nodulation response across these 15 isolates.

## Results

A total of 46 bacterial isolates were recovered from common bean nodules and identified as rhizobia considering their morphological and biochemical traits. Most of the isolates displayed fast growth, forming elevated colonies with appearances ranging from creamy white to translucent white, accompanied by slime production on YMA medium. All isolates grew on YMA supplemented with Congo Red, producing colonies that varied in color from pink to translucent. Additionally, most isolates exhibited acid production when cultured on BTB medium. Rhizobial isolates exhibited variation in their ability to grow at different temperatures and NaCl concentrations. Eleven isolates (24.4%) grew at 37 °C and four isolates (7.4%) grew at 40 °C, whereas seven isolates (15.6%) grew at 1% NaCl and three isolates (6.7%) grew at 2% NaCl (S2 Table).

### Phylogenetic analysis of *recA* gene

All 46 isolates recovered from *P. vulgaris* nodules, along with two reference strains (UMR1899 and UMR1632), were identified as *Rhizobium* based on partial *recA* gene sequence analysis. The phylogenetic tree constructed from these sequences grouped the isolates into nine distinct clusters: R1 (26 isolates, 56.5%), R2 (1 isolate, 2.2%), R3 (2 isolates, 4.3%), R4 (1 isolate, 2.2%), R5 (1 isolate, 2.2%), R6 (3 isolates, 6.5%), R7 (4 isolates, 8.7%), R8 (2 isolates, 4.3%) and R9 (6 isolates, 13.0%) (Fig 2).

Nucleotide similarity among the sequences, as well as with reference type strains ranged from 82.4% to 100% (S1 Fig). Isolates in group R1 clustered with *R. ecuadorense* CNPSo 671$^T$ and other strains of the same species (CNPSo 676, CNPSo 683, CNPSo 672, CNPSo 670). Groups R2, R3, R4, and R5 did not cluster with any reference type strains or sequences available in the NCBI database. Cluster R6 included isolates closely related to *R. leguminosarum* bv. trifolii BR 268$^T$, while group R7 clustered isolates related with *R. phaseoli* ATCC 14482T and the non–type strain CNPSo669. In cluster R8, isolates grouped with *R. etli* CFN 42$^T$, and those in cluster R9, with *R. tropici* CIAT 899$^T$. Overall, the phylogenetic analysis revealed two major lineages: **i)** clusters R1–R8 grouped into the *R. ecuadorense*/*R. etli*/ *R. leguminosarum*/ *R. phaseoli* lineage, and **ii)** cluster R9, grouped within the *R. tropici* lineage. Based on these results, 19 representative isolates were selected from distinct phylogenetic clusters for *16S rRNA* gene amplification, and multilocus sequence analysis.

### *16S rRNA* gene analysis

The ribosomal RNA (rRNA) gene was successfully amplified in 16 of 19 plus two representative strains. Three isolates failed *16S rRNA* amplification, leaving 16 isolates plus two reference strains for ribosomal analysis. A region of 672 base pair region, encompassing V1, V2, V3 and V4 hypervariable regions, was used for alignment. Phylogenetic analysis of the *16S rRNA* sequences confirmed that all isolates belonged to the genus *Rhizobium*. As expected, the conserved nature of *16S rRNA* gene within the genus *Rhizobium* limited its ability to resolve closely related species [41].

### Phylogenetic analysis of concatenated housekeeping *recA, glnII* and *dnaK*

Nineteen (19) representative isolates, including control strains UMR1899 and UMR1632, successfully amplified all three housekeeping genes *recA*, *glnII*, and *dnaK*. Phylogenetic analysis, based on concatenated sequences of the housekeeping genes *recA, glnII*, and *dnaK* (1,115 bp) from 19 fully characterized isolates, grouped the isolates into four distinct clusters, labeled C1−C4 (Fig 3).

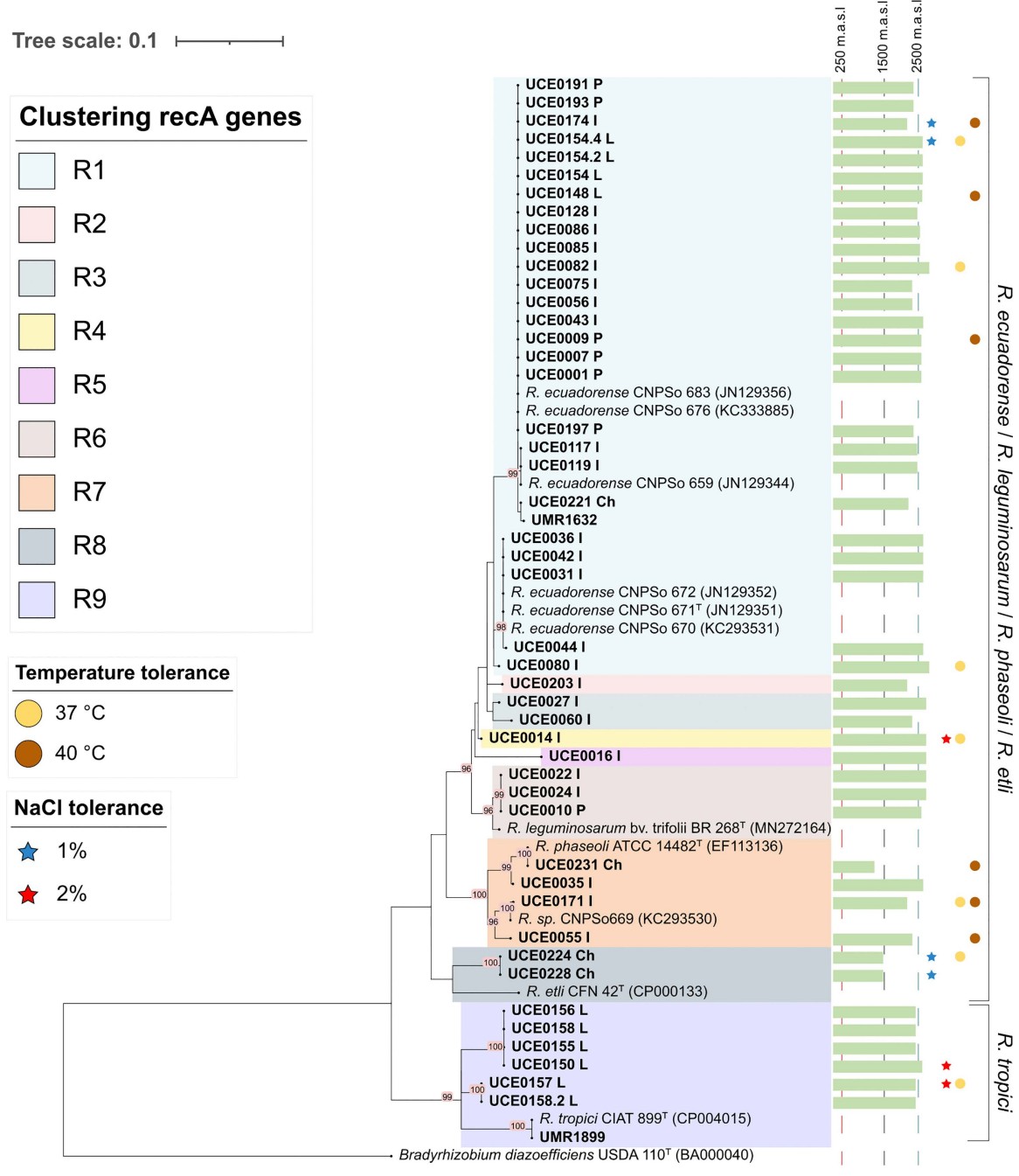

**Fig 2. Phylogenetic tree constructed from partial *recA* gene sequences of 46 isolates, including two reference strains (UMR1899 and UMR1632) and type/reference strains (indicated with superscript "T" following the strain code).** The analysis was performed using the Maximum Likelihood method based on the Tamura 3–parameter model in MEGA X. A total of 395 positions were included in the final dataset. Bootstrap values (1000 replicates) above 70% are represented at the nodes. The scale bar represents the number of substitutions per site. The green bars indicate the altitude range where samples were collected, spanning from 1200 to 2824 meters above sea level. Isolates were also assessed for salt tolerance (indicated by filled stars) and temperature tolerance (indicated by filled circles).

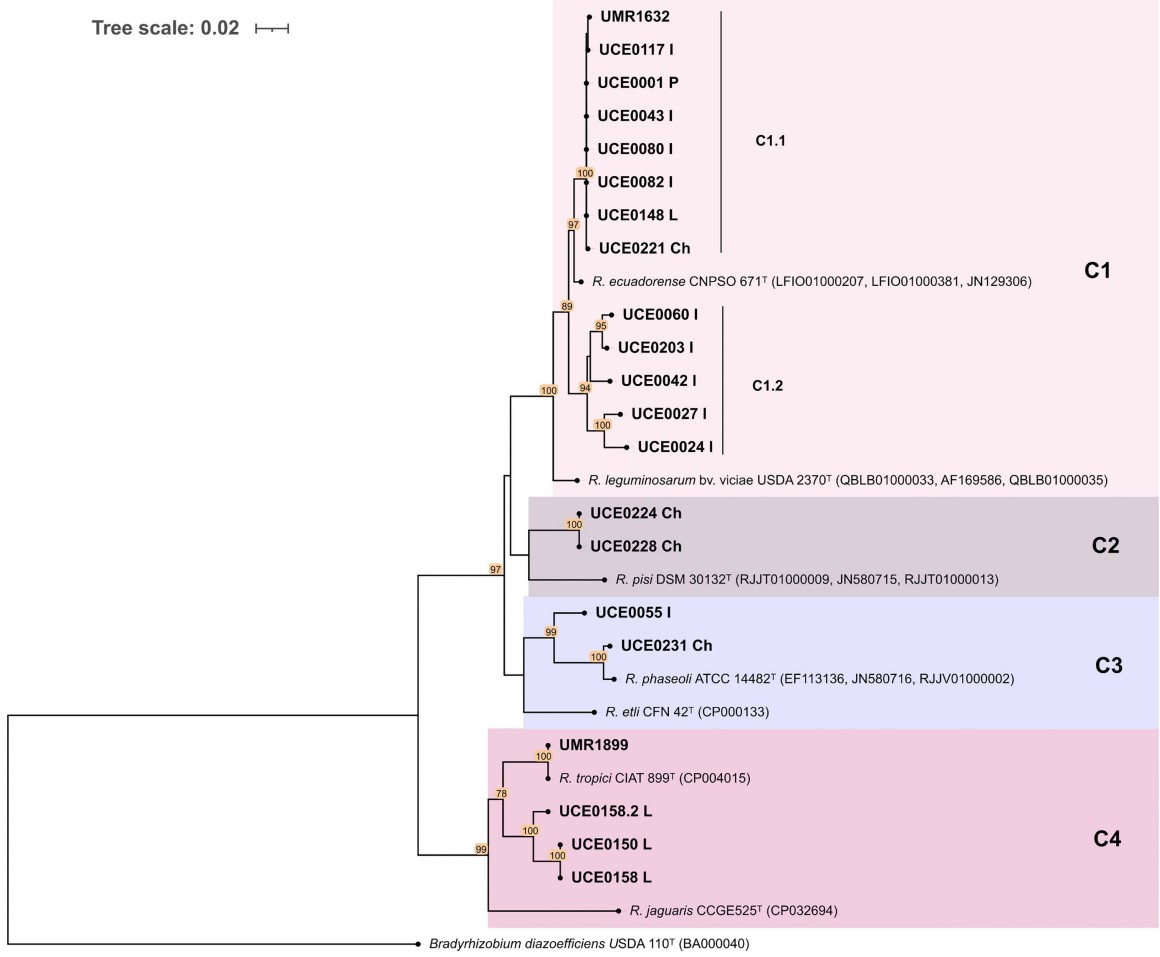

**Fig 3. Phylogenetic tree constructed from partial concatenated sequences of *recA*, *glnII* and *dnaK* genes including two reference strains (UMR1899 and UMR1632) and type/reference strains (indicated with superscript "T" following the strain code).** The analysis was performed using the Maximum Likelihood method based on the Tamura 3–parameter model in MEGA X. A total of 1115 positions were included in the final dataset. Bootstrap values (1000 replicates) above 70% are represented at the nodes. The scale bar represents the number of substitutions per site.

Cluster C1 included over half of the isolates, and was divided into two subclusters: C1.1 and C1.2. Within C1.1, nucleotide similarity among isolates ranged between 99.8% and 100% and with the reference strain *R. ecuadorense* CNPSo 671$^T$ ranged between 98.8% and 98.9%, and from 96.6% to 96.7% with *R. leguminosarum* bv. viciae USDA 2370$^T$. Within C1.2, isolates showed nucleotide similarities of 96.5% to 99.2% with each other, from 96.2% to 97.8% with *R. ecuadorense* CNPSo 671$^T$, and from 95.1% to 95.7% with *R. leguminosarum* bv. viciae USDA 2370$^T$.

Cluster C2 included isolates UCE0228 and UCE0224, which displayed 100% of nucleotide similarity with each other. These isolates shared 92.9% nucleotide similarity with the closest strain, *R. pisi* DSM 30132$^T$.

Cluster C3 comprised isolates UCE0055 and UCE0231, which showed a pairwise nucleotide similarity of 95.2% with each other. Comparative analysis with reference strains indicated that these isolates shared nucleotide similarities of 94.9%–99.0% with *R. phaseoli* ATCC 14482$^T$, and 91.6%–93.3% with *R. etli* CFN 42$^T$.

Cluster C4 comprised isolates UCE0150, UCE0158, and UCE0158.2, showing nucleotide similarities of 97.6%–100% among themselves, 94.1%–94.9% with *R. tropici* CIAT 899$^T$, and 89.6%–90.2% with *R. jaguaris* CCGE525$^T$. Results are available in S2 Fig and also summarized in Table 1.

**Table 1. Pairwise percentage nucleotide similarities of the concatenated housekeeping genes *recA*, *glnII* and *dnaK* used in multilocus sequence analysis (MLSA).**

| Cluster name | Sub–cluster name | UCE strains | Nucleotide similarities among local isolates (range %) | Range of nucleotide similarities with the closest type strain (%) |
|---|---|---|---|---|
| C1 | C1.1 | UCE0001 P<br>UCE0043 I<br>UCE0080 I<br>UCE0082 I<br>UCE0117 I<br>UCE0148 L<br>UCE0221 Ch | 99.8%–100% | *R. ecuadorense* CNPSo 671$^T$ (98.8%–98.9%)<br>*R. leguminosarum* bv. viciae USDA 2370$^T$ (96.6%–96.7%) |
|  | C1.2 | UCE0024 I<br>UCE0027 I<br>UCE0042 I<br>UCE0060 I<br>UCE0203 I | 96.5%–99.2% | *R. ecuadorense* CNPSo 671$^T$ (96.2%–97.8%)<br>*R. leguminosarum* bv. viciae USDA 2370$^T$ (95.1%–95.7%) |
| C2 | – | UCE0224 Ch<br>UCE0228 Ch | 100% | *R. pisi* DSM 30132$^T$ (92.9%) |
| C3 | – | UCE0055 I<br>UCE0231 Ch | 95.2%. | *R. phaseoli* ATCC 14482$^T$ (94.9%–99.0%)<br>*R. etli* CFN 42$^T$ (91.6%–93.3%) |
| C4 | – | UCE0158.2 L<br>UCE0150 L<br>UCE0158 L | 97.6%–100% | *R. tropici* CIAT 899$^T$ (94.1%–94.9%)<br>*R. jaguaris* CCGE525$^T$ (89.6%–90.2%) |

## Authentication of nodulation

Of the 46 rhizobial isolates evaluated, including the reference strains UMR1899 and UMR1632, 45 induced nodule formation on both common bean (*P. vulgaris*) varieties, Canario and Centenario. The sole exception was isolate UCE0148, which failed to produce nodules because the inoculated seedlings did not survive the greenhouse trial. The three Chimborazo–derived isolates were excluded from this assay for the reasons described above. In the Canario variety, the average nodulation per inoculated bacterial isolate was 47 nodules per plant, with significant variability observed among isolates, some produced up to 200 nodules, while others showed minimal nodulation, producing as few as one nodule per plant. Notably, the commercial inoculant UMR1899 yielded an average of 33 nodules, underperforming compared to several native isolates that demonstrated superior nodulation efficiency (S4 Table). Similarly, in the bean Centenario variety, bacterial inoculation resulted in an average of 49 nodules per plant with nodulation ranging from as few as one nodule to as many as 300 nodules per plant (S4 Table). The commercial inoculant UMR1899 performed better in this variety, inducing an average of 127 nodules; however, several native strains outperformed UMR1899, achieving higher and more efficient nodulation (S4 Table). Table 2 presents the performance data on nodule production, expressed as the mean number of nodules per plant ± standard deviation, for 15 molecularly characterized isolates and two reference strains.

Fig 4B and 4C include a phenotypic analysis of these 15 isolates, revealing geographical differences in their performance. However, these variations were not associated with the geographic distance among the isolates (Fig 4A), suggesting that other factors, such as edaphoclimatic conditions or host–specific adaptations, may play a more significant role in determining their effectiveness. This finding underscores the complexity of *Rhizobium*–environment interactions and highlights the importance of considering both genetic and environmental factors when selecting strains for inoculation.

**Table 2.** Nodulation performance of common bean varieties Canario (Ca) and Centenario (Ce) following inoculated with 17 rhizobia isolates (15 representative isolates + reference strains). Data are present as mean($\bar{x}$) and standard deviation (Sd). NA values indicate seedlings inoculated with this isolate did not survive the greenhouse trial. Chimborazo's strains were not included by reasons previously mentioned.

| | Isolate Collection Number | Bean variety Canario (Ca) | | Bean variety Centenario (Ce) | |
|---|---|---|---|---|---|
| | | $\bar{x}$ | Sd | $\bar{x}$ | Sd |
| 1 | UCE0001 | 11.7 | 3.1 | 82.2 | 6.7 |
| 2 | UCE0024 | 69.0 | 19.1 | 16.7 | 4.1 |
| 3 | UCE0027 | 1.3 | 0.6 | 88.7 | 22.5 |
| 4 | UCE0042 | 1.6 | 1.2 | 15.7 | 16.8 |
| 5 | UCE0043 | 57.0 | 12.5 | 11.3 | 11.1 |
| 6 | UCE0055 | 17.3 | 4.7 | 1.0 | 0.0 |
| 7 | UCE0060 | 17.7 | 9.3 | 17.7 | 9.3 |
| 8 | UCE0080 | 5.7 | 5.0 | 36.3 | 12.3 |
| 9 | UCE0082 | 100.7 | 10.6 | 70.3 | 2.5 |
| 10 | UCE0117 | 33.7 | 9.3 | 67.7 | 10.3 |
| 11 | UCE0148 | NA | NA | NA | NA |
| 12 | UCE0150 | 38.0 | 18.1 | 24.0 | 14.1 |
| 13 | UCE0158 | 79.0 | 14.5 | 7.7 | 6.1 |
| 14 | UCE0158.2 | 9.0 | 7.0 | 131.3 | 41.0 |
| 15 | UCE0203 | 5.7 | 8.1 | 31.3 | 41.0 |
| 16 | UMR1632 | 30.3 | 11.0 | 12.0 | 11.0 |
| 17 | UMR1899 | 33.3 | 11.1 | 127.7 | 34.6 |

Based on the evaluation of standardized quantitative variables, as illustrated in the heatmap in Fig 4B, several isolates from Imbabura province exhibited moderate nodulation effectiveness on the Canario bean variety (UCE0024, UCE0043, UCE0224, and UCE0082). Among these, isolate UCE0082 demonstrated the highest performance. Additionally, isolate UCE0158 from Loja province also showed significant nodulation efficacy in the same bean variety. In the case of the Centenario bean variety, no evidence of a differentiated response was observed with respect to the geographical origin of the isolates. Isolate UCE0060 from Imbabura exhibited good performance, while UCE0158.2 from Loja displayed the highest nodulation efficacy (Fig 4C). Differences were observed among bean varieties, isolates, and in the interaction between bean variety and rhizobial isolates, suggesting a local adaptability of rhizobial isolates to specific cultivars during nodulation authentication. Notably, isolates other than the 17 presented in Table 2 demonstrated enhanced nodulation capacity which is detailed in S4 Table but not shown in Fig 4B and 4C.

## Discussion

Foundational studies identified *R. ecuadorense* as a novel nitrogen–fixing symbiont phylogenetically positioned within the *R. phaseoli*/*R. etli*/*R. leguminosarum* clade but genomically and phenotypically distinct, which justifies its classification as a novel species, exhibiting <94.2% nucleotide identity in MLSA and <47% DNA–DNA hybridization with related taxa [21,26,28,42]. This divergence, validated by Ribeiro et al. [28,42], underscores its evolutionary uniqueness and adaptation to the Andean highlands of Ecuador, where host–mediated selection and biogeographic barriers likely drive lineage diversification. Despite its ecological significance, comprehensive analyses of rhizobial diversity remain limited, particularly in the Ecuador–Peru region, a recognized genetic reservoir for *P. vulgaris* and one of its primary centers of diversification,

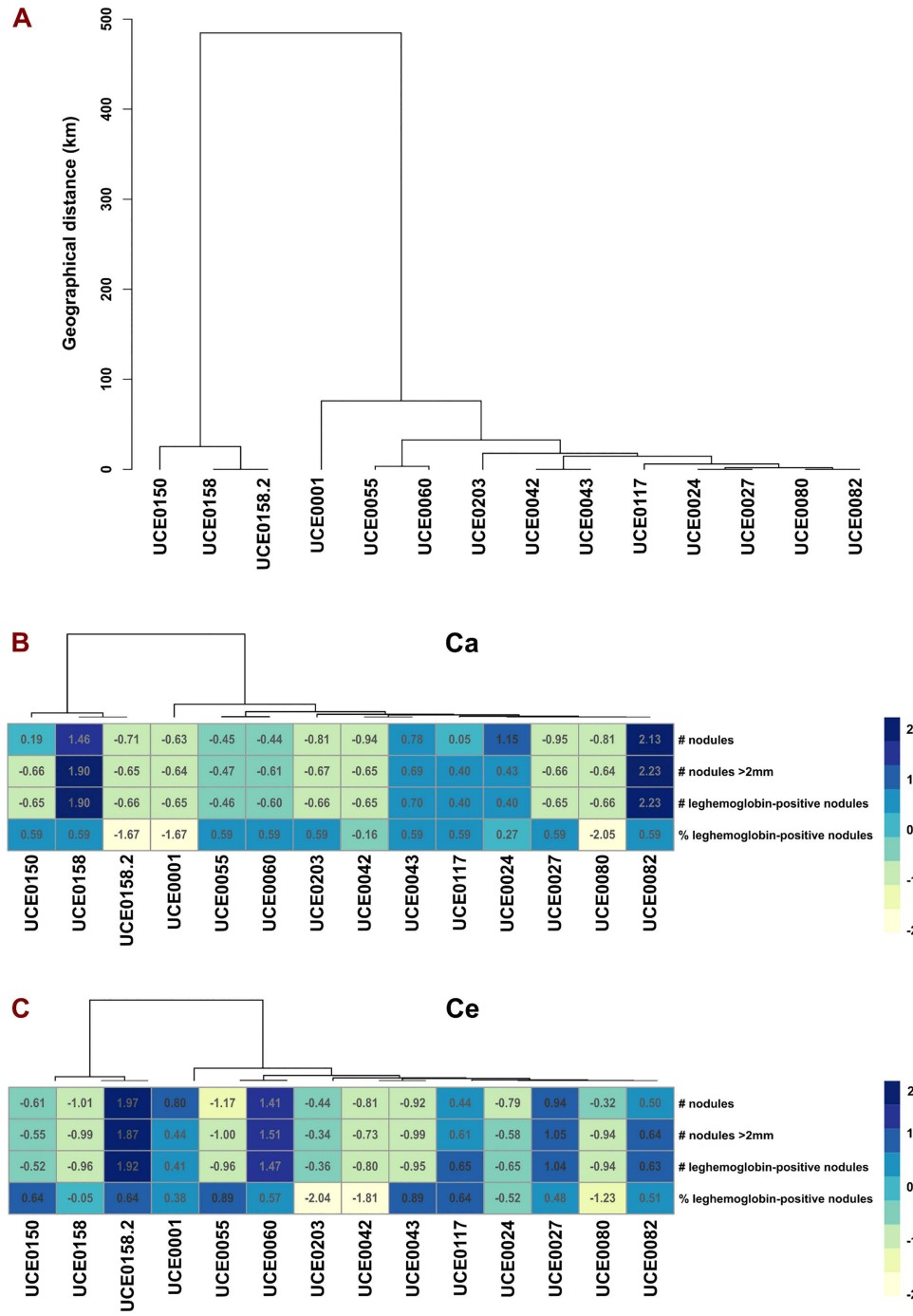

**Fig 4. Geographic clustering and standardized nodulation responses of rhizobial isolates. (A)** Hierarchical clustering dendrogram of pairwise geographic distances (km) among 14/15 representative rhizobial isolates collected from three Ecuadorian province: Imbabura, Pichincha, and Loja. Isolates from Chimborazo were excluded due to unavailability and reference strains were excluded because precise collection coordinates were unavailable. Isolate UCE0148 was omitted due to greenhouse assay failure. Loja isolates (UCE0150, UCE0158, and UCE0158.2) form a distinct cluster, UCE0001 (Pichincha) occupies an intermediate position, and the Imbabura isolates (UCE0055, UCE0060, UCE0203, UCE0042, UCE0043, UCE0117, UCE0024, UCE0027, UCE0080, and UCE0082) cluster tightly, showing tight geographic proximity. **(B, C)** Heatmaps of standardized nodulation metrics for each isolate on two *Phaseolus vulgaris* varieties: Canario (Ca) and Centenario (Ce). Metric include: (i) total nodules count per plant, (ii) Count of nodules > 2 mm, (iii) number of leghemoglobin–positive nodules, and (iv) percentage (%) of leghemoglobin-positive nodules. Color scale ranges from pale yellow (lowest standardized value) to dark blue (highest). Data represent the mean of three biological replicates and are reported in supplementary information S5 Table.

where coevolutionary dynamics between *P. vulgaris* and its microsymbionts have shaped their speciation and distribution [10,19,43,44]. Previous studies have shown that across the Americas *P. vulgaris* predominantly associates with bacteria of the *R. etli* clade as its primary nodulating symbionts [19,21,25,26]. However, *R. ecuadorense*, as a novel species, has been reported to be prevalent in northern and central Ecuador [28], suggesting a unique coevolutionary path in this Andean center of diversification.

To advance our understanding of rhizobial diversity associated with common bean cultivars in the Ecuadorian Andes, we conducted a phylogenetic analysis of 46 isolates using partial *recA* gene sequences, a marker widely validated for rhizobial identification [45–47]. The phylogeny of the *recA* gene revealed nine clusters (R1–R9) distributed into two main clusters: the *R. ecuadorense*/*R. leguminosarum*/*R. phaseoli*/*R. etli* cluster and the *R. tropici* cluster (Fig 3). These results partially align with those of Ribeiro *et al*. [21,26], may be attributable to differences in sample size between studies. *16S rRNA* sequencing of 19 representative isolates (plus the two reference strains) confirmed genus–level assignment for 16 of the 19 isolates [48–50]; and reference strains. Amplification failed for three strains, potentially due to primer–template mismatches or other technical issues.

MLSA targeting the *recA*, *glnII*, and *dnaK* genes categorized the 19 selected representative *Rhizobium* strains into four distinct clusters (C1–C4). This MLSA grouping was concordant with the phylogenetic relationships previously established using *recA* sequence analysis, a reliable marker for delineating closely related rhizobial species [47,51].

Cluster C1 was dominated by isolates closely related to *R. ecuadorense* CNPSo 671$^T$, which were prevalent in the provinces of Imbabura, Pichincha and Loja but rare in Chimborazo (Fig 3). This distribution aligns with the reported abundance in northern/central Ecuador in prior studies by Ribeiro et al. [21,28], but contrast with the absence of *R. ecuadorense* reported by Torres–Gutiérrez et al. [52] in southern Ecuador (Loja province). Such discrepancy is potentially attributable to the reliance on 16S rRNA gene analysis, which has limited resolution for distinguishing closely related *Rhizobium* species [21].

Additionally, within cluster C1, two subclusters (C1.1–C1.2) were resolved. Subcluster C1.1 exhibited >99.8% nucleotide identity with *R. ecuadorense* CNPSo 671$^T$ across the concatenated genes, a threshold often used in *Rhizobium* as a provisional species boundary but insufficient as a sole criterion [21,47,51,53,54]. In contrast, subcluster C1.2 exhibited 96.5–99.2% sequence similarity among isolates, but <97.8% similarity to its closest type strains (*R. ecuadorense* CNPSo 671 and *R. leguminosarum* USDA 2370). This observation suggests that isolates in subcluster C1.2 represent closely related taxa within a distinct lineage from the *R. ecuadorense*/-*R. leguminosarum* complex. Although phylogeny and MLSA values in our data are consistent, genomic validation through average nucleotide identity (ANI) and digital DNA–DNA hybridization (dDDH) is required to robustly delineate species boundaries.

On the other hand, cluster C2 revealed a lineage that appears taxonomically distinct. The two isolates in C2, UCE0228 and UCE0224, were identical (100% nucleotide identity) but showed no close affiliation with any known type strain, suggesting an undescribed lineage potentially endemic to the central highlands. However, expanded sampling and genome sequencing are required to assess geographic endemism.

Isolates in cluster C3 (UCE0055 and UCE0231) were positioned within the *R. etli*/-*R. phaseoli* complex. UCE0055 and UCE0231 showed nucleotide similarities of 94.9% and 99%, respectively, to *R. phaseoli*, but less than 93.3% to *R. etli,* suggesting a closer phylogenetic affinity to *R. phaseoli*. The absence of *R. etli*–related lineages and the predominance of *R. phaseoli*–associated strains within this group contrast with prior studies reporting *R. etli* as the primary symbiont in centers of *P. vulgaris* origin and diversification [19,24–26]. The absence of *R. etli* in this study, coupled with the detection of *R. phaseoli*–related lineages, suggests biogeographic or host–specific drivers of rhizobial composition in this region. These findings contrast with reports showing that *R. etli* and *R. phaseoli* predominate in Mexico, Argentina, other parts of South America, several African countries, Europe, and some Asian countries, following the introduction of common bean [13,47,51,55].

Isolates from cluster C4 were identified in samples from the province of Loja and formed a coherent group within the *R. tropici* clade, and appeared to represent a distinct lineage ([Fig 3]). These isolates, collected exclusively from Loja province, exhibited unusual phenotypic traits (white, opaque colonies, requiring at least seven days for full development, and acidification of BTB medium) that differ from typical *R. tropici* characteristics. Notably, all C4 isolates originated from soils within the Amotape–Huancabamba Zone (AHZ) in southern Ecuador, a geological well-documented biogeographic corridor extending into northern Peru [56]. The co–occurrence of this lineage with the AHZ suggests that unique environmental conditions (e.g., soil pH, organic matter, climate) may have driven divergence [57]. Genomic analysis of C4 strains is therefore required to assess their evolutionary history and local adaptations. Further investigation of environmental factors, such as soil pH, organic matter content, and other abiotic conditions, is necessary to evaluate the role of these factors in determining *Rhizobium* distribution [47,57–60].

Inoculation trials were conducted with 46 native *Rhizobium* isolates and two reference strains UMR1899 (*Rhizobium tropici* IIB CIAT 899[T)]) and UMR1632 (*Rhizobium etli* CIAT632), on two bush bean (*P. vulgaris*) varieties (Canario and Centenario). Multiple native strains outperformed the commercial inoculant UMR1899 and reference strain UMR1632 across all evaluated parameters ([Tables 2] and [S4]). These results align with prior greenhouse and field studies demonstrating superior symbiotic efficiency of native rhizobia versus commercial strains in common bean [61,62]. Notably, under controlled greenhouse conditions, the most effective isolates belonged to C1 cluster (closely related to *R. ecuadorense*) and cluster C4 (affiliated with *R. tropici*) with varietal–specific dominance patterns [Tables 2] and [S4]. Therefore, field trials are required to validate the nodulation competitiveness and symbiotic effectiveness of these elite native strains under agronomic conditions, and to sequence *nodC* genes to test the hypothesis of Aguilar et al. [19] that host variety determines rhizobial preference through diversification of symbiosis genes. In addition, future studies should incorporate direct functional assays of nitrogenase activity, such as the acetylene reduction assay, to complement nodulation data and more precisely quantify the contribution of selected isolates to biological nitrogen fixation.

## Supporting information

**S1 Table. Geographic locations, coordinates and soil type of sampling sites for *Rhizobium* isolates.** This table provides the latitude, longitude, altitude, and soil type for each of the 46 collection sites used in this study.
(DOCX)

**S2 Table. Growth tolerance of *Rhizobium* isolates to NaCl and temperature, and NCBI GenBank accession numbers for *16S rRNA*, *recA*, *glnII*, and *dnaK* sequences.** This table reports each isolate's growth tolerance at the tested NaCl (1%, 2%) (w/v) concentrations and incubation temperatures (37 °C, 40°C), and lists the corresponding NCBI GenBank accession numbers for *16S rRNA*, *recA*, *glnII*, and *dnaK*.
(DOCX)

**S3 Table. Primer sequences, PCR reaction components, and thermal cycling conditions used for amplification of target genes.** Lists primer names and sequences, reagent concentrations/volumes per PCR reaction, and the cycling parameters used for each target gene.
(DOCX)

**S4 Table. Nodulation performance of two common bean (*Phaseolus vulgaris*) varieties (Canario [Ca] and Centenario [Ce]) inoculated with 43 rhizobial isolates and two reference strains under greenhouse conditions.** For each isolate–variety combination, the table reports the mean number of nodules ($\mu$) and corresponding standard deviation ($\sigma$), based on three independent biological replicates. Nodules were scored 45 days post–inoculation. Isolates from Chimborazo were excluded due to unavailability at the moment of the greenhouse trial. In addition, an Excel (.xlsx) file will be included containing the mean ($\mu$) and standard deviation ($\sigma$) values for each of the three independent biological replicates

for the following variables: count of nodules > 2 mm, number of leghemoglobin–positive nodules, and percentage (%) of leghemoglobin-positive nodules. In yellow, isolates molecularly characterized with three housekeeping genes *recA*, *glnII* and *dnaK*.
(DOCX)

**S5 Table. Raw data of the quantitative variables of nodulation performance evaluated in two varieties of bush bean Centenario (Ce) and Canario (Ca) after being inoculated with rhizobia isolates.** Raw data from the three replications of the three independent biological replicates for the following variables: i) number of nodules, ii) count of nodules > 2 mm, iii) number of leghemoglobin–positive nodules, and iv) percentage (%) of leghemoglobin-positive nodules.
(DOCX)

**S1 Fig. Evolutionary divergence of *recA* sequences from study isolates and their closest described type strains.** Pairwise percent–identity values estimates were calculated from a 395 bp alignment of *recA* sequences to quantify the evolutionary divergence between each study isolate and its nearest described type strain. Evolutionary analyses were carried out using MEGA X; alignment and analysis parameters are detailed in the Methods.
(TIFF)

**S2 Fig. Evolutionary divergence of concatenated *recA–glnII–dnaK* sequences from study isolates and their closest described type strains.** Pairwise percent–identity estimates were calculated from a 1,115 bp concatenated alignment (*recA*, *glnII*, *dnaK*) to quantify evolutionary divergence between each study isolate and its nearest described type strain. Evolutionary analyses were performed using MEGA X software; alignment and analysis parameters are described in the Methods.
(TIF)

## Acknowledgments

The authors are grateful for the logistical and laboratory support provided by the Instituto de Investigación en Etnociencias and the Instituto de Investigación en Zoonosis (CIZ).

## Author contributions

**Conceptualization:** Andrea León–Cadena, José Ochoa, Lenin Ron–Garrido.

**Data curation:** Andrea León–Cadena, Janine Jiménez–Parra, Michelle Avalos–Loayza, Pamela Murillo, Lenin Ron–Garrido.

**Formal analysis:** Andrea León–Cadena, Henry D. Naranjo, Lenin Ron–Garrido.

**Funding acquisition:** Andrea León–Cadena, José Ochoa, Ángel Murillo, Juan Cadena–Villota, Lenin Ron–Garrido.

**Investigation:** Andrea León–Cadena, Janine Jiménez–Parra, Gustavo Bernal, Lenin Ron–Garrido.

**Methodology:** Andrea León–Cadena, Henry D. Naranjo, José Ochoa, Ángel Murillo, Gustavo Bernal, Lenin Ron–Garrido.

**Project administration:** Andrea León–Cadena, Juan Cadena–Villota, Lenin Ron–Garrido.

**Resources:** Juan Cadena–Villota.

**Software:** Lenin Ron–Garrido.

**Supervision:** Lenin Ron–Garrido.

**Validation:** Andrea León–Cadena, Henry D. Naranjo, Lenin Ron–Garrido.

**Visualization:** Andrea León–Cadena, Henry D. Naranjo, Lenin Ron–Garrido.

**Writing – original draft:** Andrea León–Cadena.

**Writing – review & editing:** Andrea León–Cadena, Henry D. Naranjo, Gustavo Bernal, Lenin Ron–Garrido.

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
