## [Decision Letter · Decision Letter 0]

20 Sep 2025

Dear Dr. Ron Garrido,

Thank you for submitting your manuscript to PLOS ONE. After careful consideration, we feel that it has merit but does not fully meet PLOS ONE’s publication criteria as it currently stands. Therefore, we invite you to submit a revised version of the manuscript that addresses the points raised during the review process.

We look forward to receiving your revised manuscript.

Kind regards,

Ying Ma, Ph.D.

Academic Editor

PLOS ONE

Journal Requirements:

4. Thank you for stating the following in the Acknowledgments Section of your manuscript: [This study was supported by the Universidad Central del Ecuador through the Convocatoria de Proyectos de Investigación Senior UCE-2019 (DI-CONV-2019-033). Additional support in logistics and laboratory infrastructure was provided by the Instituto de Investigación en Etnociencias and the Instituto de Investigación en Zoonosis (CIZ).]

Please remove any funding-related text from the manuscript and let us know how you would like to update your Funding Statement. Currently, your Funding Statement reads as follows: [This study was supported by the Universidad Central del Ecuador through the Convocatoria de Proyectos de Investigación Senior UCE-2019 (DI-CONV-2019-033). Additional support in logistics and laboratory infrastructure was provided by the Instituto de Investigación en Etnociencias and the Instituto de Investigación en Zoonosis (CIZ). The funders had no role in study design, data collection and analysis, decision to publish, or preparation of the manuscript.]

7. We note that Figure 1 in your submission contain [map/satellite] images which may be copyrighted. All PLOS content is published under the Creative Commons Attribution License (CC BY 4.0), which means that the manuscript, images, and Supporting Information files will be freely available online, and any third party is permitted to access, download, copy, distribute, and use these materials in any way, even commercially, with proper attribution. For these reasons, we cannot publish previously copyrighted maps or satellite images created using proprietary data, such as Google software (Google Maps, Street View, and Earth). For more information, see our copyright guidelines: http://journals.plos.org/plosone/s/licenses-and-copyright.

Reviewers' comments:

Reviewer's Responses to Questions

**Comments to the Author**

1. Is the manuscript technically sound, and do the data support the conclusions?

Reviewer #1: Yes

Reviewer #2: Yes

2. Has the statistical analysis been performed appropriately and rigorously?

Reviewer #1: Yes

Reviewer #2: Yes

3. Have the authors made all data underlying the findings in their manuscript fully available?

Reviewer #1: Yes

Reviewer #2: No

4. Is the manuscript presented in an intelligible fashion and written in standard English?

Reviewer #1: Yes

Reviewer #2: Yes

Reviewer #1: The article has been significatly improved after revition and all my previous concern has been addres to my satisfaction, hence i recomed it for publication. i recomended a english revition, and is pisivilit to nitrogenace activity medition

Reviewer #2: Manuscript contains very good wedge of research in order to characterize and understand the diversity of Rhizobium symbionts of the Ecuadorian region and its role in sustainable agriculture. The method used for the study multilocous sequence analysis (MLSA) use three house housekeeping genes is extremely suitable for these kind of studies. Though every care was taken in drafting the manuscript but at some pint it need to refined/corrected:

1. Figure 2 is not readable the quality of the figure need to be enhanced for the better understanding and interpretation of the results.

2. Figure 4 is also blurred and non readable, it quite tough to make any interpretation out of it . its visibility need to enhanced as it is mentioned several time in the results and created lots of hindrance in understanding and making any clear outcome of the pictures.

3. The table mentioned in the line number 345 & 347, i.e., Table No 4 is missing , make sure it is uploaded with the manuscript to know the variation in number of nodules due to different isolates.

**Do you want your identity to be public for this peer review?** For information about this choice, including consent withdrawal, please see our Privacy Policy

Reviewer #1: **Yes: ** Alfonso Leija Salas

Reviewer #2: **Yes: ** Rajesh Kumar Singh

---

## [Author Response · Author response to Decision Letter 1]

3 Dec 2025

All the comments and suggestions are indicated in the rebuttal letter.

---

## [Editor Report · Decision Letter 1]

11 Dec 2025

Updating the description of Rhizobium diversity associated with common bean cultivars in the Ecuadorian Andes: a phylogenetic and functional perspective.

PONE-D-25-33605R1

Dear Dr. Garrido,

We’re pleased to inform you that your manuscript has been judged scientifically suitable for publication and will be formally accepted for publication once it meets all outstanding technical requirements.

Kind regards,

Ying Ma, Ph.D.

Academic Editor

PLOS One
---

## [Editor Report · Acceptance letter]

PONE-D-25-33605R1

PLOS One

Dear Dr. Ron Garrido,

I'm pleased to inform you that your manuscript has been deemed suitable for publication in PLOS One. Congratulations! Your manuscript is now being handed over to our production team.

Kind regards,

on behalf of

Dr. Ying Ma

Academic Editor

PLOS One